# Tailor-made biochar systems: Interdisciplinary evaluations of ecosystem services and farmer livelihoods in tropical agro-ecosystems

**Severin-Luca Bellè** [1], **Jean Riotte**[2,3], **Norman Backhaus**[1,4], **Muddu Sekhar**[3,5], **Pascal Jouquet**[3,6], **Samuel Abiven**[7,8] *

**1** Department of Geography, University of Zurich, Zurich, Switzerland, **2** Géosciences Environnement Toulouse, Université Paul-Sabatier, IRD, CNRS, Toulouse, France, **3** Indo-French Cell for Water Science, Indian Institute of Science, Bangalore, Karnataka, India, **4** University Research Priority Programme (URPP) Global Change and Biodiversity, University of Zurich, Zurich, Switzerland, **5** Department of Civil Engineering, Indian Institute of Science, Bangalore, Karnataka, India, **6** Institut d'écologie et des Sciences de l'environnement, IESS-Paris UMR Sorbonne Université, UPEC, CNRS, IRD, INRAe, FEST Team, Bondy, France, **7** Département de Géosciences, Laboratoire de Géologie, CNRS – École Normale Supérieure, PSL University, Institut Pierre Simon Laplace, Paris, France, **8** CEREEP-Ecotron Ile De France, ENS, CNRS, PSL University, St-Pierre-lès-Nemours, France

* abiven@biotite.ens.fr

**Data Availability Statement:** All relevant data available from Zenodo: DOI 10.5281/zenodo.5769577.

## Abstract

Organic matter management is key to sustain ecosystem services provided by soils. However, it is rarely considered in a holistic view, considering local resources, agro-environmental effects and harmonization with farmers' needs. Organic inputs, like compost and biochar, could represent a sustainable solution to massive current challenges associated to the intensification of agriculture, in particular for tropical regions. Here we assess the potential of agricultural residues as a resource for farmer communities in southwestern India to reduce their dependency on external inputs and sustain ecosystem services. We propose a novel joint evaluation of farmers' aspirations together with agro-environmental effects of organic inputs on soils. Our soil quality evaluation showed that biochar alone or with compost did not improve unilaterally soils in the tropics (Anthroposol, Ferralsol and Vertisol). Many organic inputs led to an initial decrease in water-holding capacities of control soils (-27.3%: coconut shell biochar with compost on Anthroposol). Responses to organic matter inputs for carbon were strongest for Ferralsols (+33.4% with rice husk biochar), and mostly positive for Anthroposols and Vertisols (+12.5% to +13.8% respectively). Soil pH responses were surprisingly negative for Ferralsols and only positive if biochar was applied alone (between -5.6% to +1.9%). For Anthroposols and Vertisols, highest increases were achieved with rice husk biochar + vermicomposts (+7.2% and +5.2% respectively). Our socio-economic evaluation showed that farmers with a stronger economical position showed greater interest towards technology like biochar (factor 1.3 to 1.6 higher for farmers cultivating Anthroposols and/or Vertisols compared to Ferralsols), while poorer farmers more skepticism, which may lead to an increased economical gap within rural communities if technologies are not implemented with long-term guidance. These results advocate for an interdisciplinary evaluation of agricultural technology prior to its implementation as a development tool in the field.

**Funding:** This work was supported by the Swiss National Science Foundation (https://www.snf.ch/en) grant number 200021_178768 awarded to SA. The funders had no role in study design, data collection and analysis, decision to publish, or preparation of the manuscript.

**Competing interests:** The authors have declared that no competing interests exist.

# Introduction

Agricultural intensification over the last decades has contributed substantially to the decrease of ecosystem services provided by soils [1, 2]. The (over)use of synthetic inputs in agricultural production has unbalanced biogeochemical cycles, depleted natural resource pools, contaminated water bodies, polluted and degraded the arable soils as the backbone of agriculture and ultimately increased the dependency of farmers [2–4]. In India, the liberalization of agriculture during the green revolution in the 1960ies induced a trend towards cash crop production [5, 6] and contributed to major ecological as well as socio-economic crises [7]. The reduction of soil fertility, and accordingly decreasing crop yields, has led to significant farmers' distress and indebtedness [5, 8], necessitating an urgent, yet possible [9] shift towards sustainable agricultural practices and simultaneous support to ecosystem services essential for food production and farmers livelihoods [10, 11]. Organic matter management (OMM) can regulate essential biochemical and physical processes in soils and sufficient, long-term amounts of soil organic matter (SOM) is precondition for soils to provide the ecosystem services needed for durable soil fertility and agricultural production [12–14]. Availability of organic matter (OM; crop residues, animal excreta) in India can be between 320 up to 840 Million tons annually [9, 15], which equals on average 2.5 tons OM per hectare (considering a total cropping area of 180.9 million hectares [9]). The surplus OM (i.e. OM not used for any other domestic purposes) is between 25–72%, including major crops grown in the study area in Karnataka (sugarcane bagasse, wheat, coconut shell, rice straw, banana) [16, 17].

Agriculture in tropical regions is highly relevant regarding food production and security at the global scale [18]. However, the tropics are more vulnerable to global changes such as climate and land-use change across climate zones of the Earth [4]. The state of Karnataka, mostly located in sub-humid to semi-arid climates in the southern part of the Indian peninsula, is among the most vulnerable Indian states to climate change [19]. Major parts are prone to drought and soil degradation, which are only two of the challenges in relation to climate and land-use changes observed in many places in the tropics [19, 20]. In many parts of India, especially in the South, irrigation and consequently agricultural production depends to a large extent on monsoonal rainfall patterns and groundwater irrigation [19, 21]. In addition, soils are highly depleted in SOM, because of conditions favorable to high export of carbon through crop production, limited inputs of OM to soils [22, 23]; and high mineralization rates [18]. These soils are subject to leaching of major macro- and micronutrients, and reduction in aggregate stability [24–26], resulting in a limited potential to provide essential ecosystem services [1].

OM inputs are also key for securing agricultural sustainability of rural livelihoods through appropriate income generation [11] and reducing dependency on external inputs [4, 27]. Adaptive OMM to site-specific agro-ecosystems, which identifies and embeds traditional (local) knowledge (defined as "*tacit and explicit knowledge possessed and used by people who share the same culture*" according to the Oxford Dictionary of Human Geography ([28])) and agricultural practices of rural communities, is critical to be a successful solution to the current challenges as it include site-specific adaptation barriers for farmer communities. Tailor-made OMM considers social issues such as traditional perception and livelihood practices of farmers, autonomy and participation concerns, knowledge dissemination in existing networks and the farmers' need satisfaction [29]. Barriers also refer to site-specific economical concerns like agricultural residue availability [11, 29] and resource competition for different uses of OM [23].

Among existing OMM techniques, composting and vermicomposting (utilizing earthworms to digest pre-composted OM), involving a process of aerobic or anaerobic degradation

of mostly lingo-cellulosic biomass and/or animal excreta [30], are already perceived as valuable methods of converting OM into soil amendments [31, 32] and are applied by rural farmer communities in India [33]. Biochar, a carbonaceous product of intended heating of OM under the absence or low levels of oxygen and temperatures above 200˚C ("pyrolysis") [34], has been shown to have multifaceted effects on soil properties including changes in soil pH, porosity, water-retention potential and plant-available nutrients [35–38]. Both composts and biochar have been proposed to increase soil fertility and agricultural productivity, while simultaneously reducing the environmental impact of agriculture and socio-economic dependency of farmers [39, 40]. Next generation of OM inputs, like biochar-based fertilizers, consider the complementarity of inputs combined together. In this case, inputs with synergic and antagonistic properties are added together to the soil, for example biochar and compost, combining amendment (physico-chemical capacity of biochar) and fertilizers (compost) [38, 41].

Besides the prerequisite of a certain financial capital, availability of agricultural residues, knowledge and technical skills, biochar is considered to be a sustainable tool for helping rural farmer communities managing agricultural residues and reducing their dependency on external inputs [12, 42, 43]. Adaptation of biochar technology to specific agro-ecological and socio-economic settings, inclusion of traditional (local) knowledge and practices of rural farmer communities, and resource competition with domestic usage of OM are further concerns that need to be addressed when implementing tailor-made biochar systems to any site-specific context [3, 42, 43]. In this regard, biochar-based fertilizer can be a solution to both environmental as well as socio-economic concerns [38].

The challenges described above call for an agriculture that fulfills in the same time ecological, socio-economic as well as political aspects of sustainability [11, 44]. The benefits of OMM according to soil types and agro-ecosystems as well as the site-specific, socio-economic barriers are currently studied separately, in particular regarding applicability and alignment with the agricultural sector [29, 45]. However, there is an urgent need to connect these evaluations in an interdisciplinary effort across multiple scientific communities from the natural and social sciences in order to develop and implement innovative cropping systems adaptive to specific agro-ecosystems that improve rural farmers' livelihoods in parallel to soil ecosystem services [46–48]. We propose a novel, joint assessment of soil quality and farmers' aspiration in order to evaluate the conformity of OMM to existing rural farming communities by using a variety of complementary methodologies rooted in human geography and the soil sciences. Our interdisciplinary approach could be seen as a first practical example on how to link agro-ecological and socio-economic questions in agricultural research that can form the basis for future in-depth, long-term field trials on OMM together with local farmer communities in tropical regions [46, 47]. This could ultimately help to develop sustainable agricultural technologies for the development or local agricultural and industrial sectors. We led in parallel qualitative interviews and a soil manipulative study to evaluate the potentials of OM inputs ((vermi-) composts (derived from OM such as crop residues, cow dung, leaves, etc.), biochar (derived from coconut shell and rice husk) or next generation combination of inputs) to common tropical soil types from southwestern India (Anthroposol, Ferralsol and Vertisol). To do so, we estimated effect sizes of soil quality indicators, selected from a list of commonly used parameters, such as water-holding capacity (WHC), total carbon, total nitrogen (TC and TN) and pH after OM application [14]. Then, we evaluated farmers' aspiration through in-depth, qualitative interviews based on a predefined topic guide and by selecting farmers cultivating the same soils as those experiments took place. The study was conducted in the cultivated watershed of Berambadi in southwestern India, which is part of the Kabini CZO and SNO M-TROPICS program [49].

## Material and methods

### Case study: The Berambadi watershed in southwestern Karnataka (India)

Research was carried out in the Berambadi watershed (11˚43'00" to 11˚48'00" N, 76˚31'00" to 76˚4"00" E) in Chamarajanagar district in southwestern Karnataka. The watershed is situated on the Deccan plateau east of the Western Ghats and represents a sub-humid tropical watershed. It belongs to the Kabini CZO and SNO M-TROPICS program, and is a collaboration between the Indian Institute of Science (IISc) and IRD [49, 50].

The geology in the southern parts of the Deccan plateau is characterized by granitic gneiss, which represent the basis of the predominantly occurring red soils (Ferralsols and chromic Luvisols) on hills/hill slopes and the black soils (Vertisols and Vertic integrades) close to the river banks [51, 52]. The topography induced by the Western Ghats mountain range results in an increasing water availability from east towards west as well as an spatially heterogeneous pattern of soil types. These natural factors, which cause a variable environment on a small scale, make the Berambadi watershed a critical geographical region for scientific observations [50].

Agriculture in the region is dictated by monsoon dynamics and groundwater tables, and crops are grown either in Kharif season (June to September) and/or Rabi Season (October to December), whereas in summer (January to May) only limited, irrigated agriculture is practiced [50, 52]. Farmers either grow perennial (turmeric, sugarcane, banana), annual (Jowar, sunflower, Ragi) or short-term (vegetables, pulses, grams) crops, mainly dependent on the irrigation type with farmers owning bore wells cultivating all year round [49, 52].

### Soil sampling and characterization

We excavated three pits for each of the three prevalent soil types cultivated by farmers (Anthroposol, Ferralsol and Vertisol) in Berambadi watershed to a depth of 20 centimeters (agricultural horizon), subsequently mixed the soil to one composite sample for each soil, which was air-dried and packed for transportation. Samples of the Anthroposol (Ferralsol where river (tank) sediments were applied [53]) were taken on the farm of F23 in Berambadi town. Samples of the Ferralsol (field along a gentle hillslope) and Vertisol (field next to the streambed) were taken on the farm of F11 and F12 in Gopalpura town respectively. The main characteristics of the studied soils are shown in Table 1.

### Organic matter inputs

Substrates were liberally provided by the University of Agricultural Sciences (UAS Bangalore, Prof. Prakash Nagabovanalli), by the Indo-French Cell for Water Sciences (Indian Institute of Science, Bangalore) and by Karthik Vermicompost and Earthworm Consultancy. Five OM inputs have been used: a compost (C1), produced in rotating drums from a mixture of cow

**Table 1. Control soil water-holding capacity (WHC), total carbon and nitrogen (TC and TN) contents, C:N ratio and pH of the Anthroposol, Ferralsol and Vertisol.** Values are means (n = 3) ± one standard deviation (in parentheses).

|  | Anthroposol | Ferralsol | Vertisol |
|---|---|---|---|
| **WHC (%)** | 14.02 ± (0.69) | 5.97 ± (0.24) | 16.38 ± (1.20) |
| **TC (%)** | 1.13 ± (0.03) | 0.45 ± (0.03) | 1.56 ± (0.05) |
| **TN (%)** | 0.119 ± (0.004) | 0.070 ± (0.006) | 0.098 ± (0.002) |
| **C:N ratio** | 9.55 ± (0.10) | 8.64 ± (0.37) | 15.89 ± (0.30) |
| **pH (-)** | 6.97 ± (0.06) | 7.10 ± (0.10) | 7.77 ± (0.11) |

**Table 2. Water-holding capacity (WHC), total carbon and nitrogen (TC and TN) contents, C:N ratio and pH of the selected biochar (B1: Coconut shell, B2: Rice husk) and compost (C1: Compost, VC1 and VC2: Vermicomposts) inputs.** Values are means (n = 3) ± one standard deviation (in parentheses). n.d. = not detected.

| | B1 | B2 | C1 | VC1 | VC2 |
|---|---|---|---|---|---|
| **WHC (%)** | 9.27 ± (0.44) | 6.27 ± (1.49) | 16.70 ± (1.36) | 18.66 ± (0.65) | 17.28 ± (0.16) |
| **TC (%)** | 85.73 ± (4.65) | 42.66 ± (0.75) | 13.04 ± (2.38) | 11.30 ± (0.40) | 14.28 ± (1.41) |
| **TN (%)** | n.d. | 0.40 ± (0.28) | 1.37 ± (0.23) | 1.34 ± (0.10) | 1.41 ± (0.19) |
| **C:N ratio** | n.d. | 138.88 ± (69.20) | 9.48 ± (0.13) | 8.42 ± (0.16) | 10.08 ± (0.57) |
| **pH (-)** | 9.07 ± (0.06) | 7.33 ± (0.06) | 6.83 ± (0.06) | 7.43 ± (0.07) | 6.73 ± (0.06) |

dung and crop residues (dried OM, leaves, etc.) where a microbial inoculum was added (UAS Bangalore), two vermicomposts (VC1: produced in heaps of cow dung and crop residues in the first phase, then filled in long solid tanks to which a earthworm culture was introduced and then covered by plastic sheets (UAS Bangalore) and VC2: produced by farmers in heaps of cow dung and organic waste (crop residues, food waste, leaves, branches) in a ratio 3:1, then filled in solid tanks along an earthworm culture (Karthik Vermicompost and Earthworm Consultancy)) and two types of biochar (coconut shell (B1), produced with the Kiln technique for charcoal purposes and rice husk (B2), produced at the UAS Bangalore, using a retort pyrolyser). Main characteristics of the OM inputs are shown in Table 2 below.

## Soil manipulative experiment

To study the effect of individual and combined OM inputs on four selected properties (WHC, TC, C:N (TN), pH) of the three studied soils (Table 1), a static incubation experiment under controlled conditions was performed over 70 days (based on [26, 54]). Soils were pre-incubated with 15% deionized water of total mass for eight days. A total of 60 grams dry-equivalent soil was put into 0.2 liter plastic cups and then thoroughly mixed with the OM. The amount of OM to be added was calculated based on approximate application rates of farmers in Berambadi watershed. The empirical material from farmer interviews shows that farmers can generally apply between 5–10 tons of OM per hectare (t ha$^{-1}$). With an approx. bulk density of 1.3 tons per cubic meter (t m$^{-3}$), the amount is calculated as followed:

$$OM\ input = Soil_{mass} \times \frac{AR_{OM}}{Soil_{depth} x Soil_{area} x Soil_{BD}} \tag{1}$$

where OM input represents the OM input for each cup (converted from tons (t) to milligrams (mg), Soil$_{mass}$ the dry-equivalent soil mass for each cup (60 g = 0.00006 t), AR$_{OM}$ the application rate of OM by farmers (we took the upper value of the range from 5–10 t ha$^{-1}$ = 10 t ha$^{-1}$), Soil$_{depth}$ the depth of the agricultural horizon (20 centimeters = 0.2 meter), Soil$_{area}$ the reference area of one hectare in square meters (1 ha = 10'000 m$^2$) and Soil$_{BD}$ represents the approximated soil bulk density (1.3 t m$^{-3}$). For single OM inputs (only biochar or only composts), this resulted in an OM input of 230 mg to the soil (0.00006t*(10t/(0.2m*10'000m$^2$*1.3 t m$^{-3}$)) = 2.3*10$^{-7}$ t = 230 mg), for combined OM inputs a mass equivalent of 115 mg of two OM (e.g. C1 + B1). The OM was carefully mixed with the first few centimeter of soil. The soil was slightly compressed and plastic cubs were put in 2 liter glass jars alongside a 20 milliliter glass vial filled with deionized water. The jars were put into an incubator and kept at 24 °C, which corresponds to the mean annual air temperature of Berambadi weather station. Each OM input for each soil was prepared in triplicates. The samples were periodically weighed to track loss of moisture and the jars were regularly opened to renew oxygen. Controls were prepared the same way without application of OM in triplicates.

## Soil analysis

After the experiment, (soil) samples were dried for 24 hours at 40 ˚C and then milled using a planetary mill (Fritsch pulverisette 5, Fritsch GmbH, Idar-Oberstein, Germany). Biochar sub-samples were further milled with a horizontal mill (Retsch MM400, Retsch GmbH, Haan, Germany). The pH was measured with a Methrom 692 pH/Ion meter (Methrom Schweiz AG, Zofingen, Switzerland) in a 1:5 soil:water solution according to ISO 10390:2021. The sieved ($< 2$ mm) samples were preliminary stirred for 30 minutes at a rate of $< 500$ rpm, then put to settle for one hour before measurement, and then the pH was measured in the soil suspension with a glass electrode. Water-holding capacity (WHC) was assessed with a pF laboratory station from ecoTech (ecoTech Umwelt-Messsysteme GmbH, Bonn, Germany). Each replicate was weighed by volume into metal rings and the material was held by a fine tissue net. Weights were noted down and subsequently put into a water bath for 24 hours until saturation occurred. Samples were placed on a suction plate and the suction tension was set for field capacity at -325 to -335 millibar (= 33 kPa). Samples were regularly weighed and measurements stopped when only minor variations occurred ($<0.5$ grams). Samples were dried for 24 hours at 105 ˚C and then weighed. The volumetric water content (or the WHC) was calculated as the difference between the weight in equilibrium at field capacity and the dry weight. Total C and N contents were measured by an Elemental Analyzer-Isotope Ratio Mass Spectrometer (EA-IRMS; Flash 2000-HT Plus, linked by Conflo IV to Delta V Plus isotope ratio mass spectrometer, Thermo Fisher Scientific, Bremen, Germany).

## Statistical analysis

We performed an analysis of variance for the full dataset derived from the incubation experiment, testing for significant differences of measured soil properties (WHC, TC, C:N, pH) between soils, OM and their interactions (two-way ANOVA, n = 3). Subsequently, we performed another analysis of variance on the measured soil properties (WHC, TC, C:N, pH) for each soil type separately, with the OM inputs as the independent variable (one-way ANOVA, n = 3). For both ANOVA's, we used Levene's test to check the assumption of homogeneity of variance (center = mean), a Shapiro-Wilk test on the ANOVA residuals to check for the assumption of normality and a Fisher's least significant difference (LSD) post-hoc test to check for significance (alpha = 0.05, p.adj. = bonferroni). Statistical analysis was carried out using R Studio 1.3.1093 (2009–2020), R Version 4.0.3 (R Core Team, 2020). All values reported represent means with one standard deviation.

While it would have been possible to use simple, descriptive statistics on the numbers derived from the coding of empirical data of farmers' interviews, we refrained from doing so because these numbers (mainly percentages) represent assigned codes to text sequences that originate from the statements of farmers during the interviews, and not measured data (see next section). However, there is an ongoing debate in qualitative research if simple statistics can be used for qualitative data [55].

## Assessment of farmers' traditional (local) knowledge, practices and aspirations—Grounded Theory Methodology

Qualitative data collection followed the order of: 1) Explorative expert interviews (n = 9) with scientists to get an overview over the agrarian context of the research area and to initialize a network to the research area [56–58], 2) Familiarization with the area of research, leading discussions with local experts (scientists, NGO-workers and private companies) were done (n = 5) and 3) In-depth, qualitative interviews with farmers (n = 29; Anthroposol: n = 13,

Ferralsol n = 10 and Vertisol n = 6) to assess traditional (local) knowledge, practices and aspirations of farmers towards OMM in the research area [59]. Interviews with farmers in the Berambadi watershed were conducted with the help of a translator. Sampling of experts and farmers for the interviews was conducted based on the logics of theoretical sampling and saturation according to Grounded Theory Methodology [60, 61].

Qualitative data analysis was continuously performed, starting during data collection, and according to Grounded Theory Methodology, an inductive, data-driven, flexible and circular research approach in social sciences [60–62]. All audio recordings from interviews were transcribed using word-by-word transcription in order to sustain as much information as possible (NCH Express Scribe Transcription Software Pro 5.90). The empirical material (transcripts) was subsequently coded and categorized according to Grounded Theory [61–63] with the coding software MAXQDA 12 (VERBI GmbH). Analysis and comparison of codes and categories was stopped when theoretical saturation occurred, i.e. additional empirical material and their conceptualization (codes) did not contribute to new insights within the created categories [61]. Results of this process were formulated in theses about the traditional (local) knowledge, practices and aspirations of farmers on OMM (including biochar applications) in the research area. Codes about specific dimensions of the research (i.e. expectations or doubts about technology) from interview data were used to calculate the amount of farmers designating a specific point of interest (i.e. number of farmer designating interest in biochar divided by all farmers (or all farmers of each soil type)).

Qualitative research was conducted according to the "Guidelines on Ethics and Safety in Fieldwork for Researchers in Human Geography" of the Department of Geography, University of Zurich (accessible online at https://www.geo.uzh.ch/dam/jcr:d546eb46-376b-4109-ae9d-719d2d400f3d/Ethics_Guidelines.docx) and the guidelines of data protection in qualitative research described in ref [64]. Farmers were informed about the aim and content of the research study and verbally asked for their consent to participate in the interview campaign. Permission to conduct and record interviews and subsequently transcribe and analyze interview data anonymously was obtained prior to each interview. Each farmer was assigned a unique, anonymous code (F1-F29).

## Results and discussion

### Effect sizes of soil quality indicators after organic matter inputs

Neither application of biochar, composts nor combined OM inputs had a clear unilateral beneficial effect on the studied soil properties of the Anthroposol, Ferralsol and Vertisol (Figs 1 and 2; Tables 3 and 4), in line with the most recent review on the effect of biochar systems on soil and plant responses [38]. In consequence, OMM results in tradeoffs for the farmer (Fig 3) [65]. Studied parameters (WHC, TC, C:N and pH) showed strong differences between soils and OM inputs, as well as their interaction (Table 3).

Almost all OM inputs led to an initial reduction of the WHC of soils after 70 days of incubation (Fig 1a and Table 4). In case of the Anthroposol (initial WHC = 14.02 ± 0.69%), except for coconut shell biochar (B1 = 13.52 ± 0.54; p > 0.05), all OM inputs resulted in significant decreases in WHC, up to –27.29 ± 0.97% compared to the control (B2+C1 = 10.12 ± 0.14%; p < 0.05). Despite low initial value (5.97 ± 0.24%—Table 1; characteristic of soils from the sub-humid tropics [51, 67]), the WHC of the Ferralsol only tends to increase with coconut shell biochar alone (B1 = 6.07 ± 0.16%; p > 0.05) or the same biochar mixed with composts (B1+C1 = 6.57 ± 0.07%, B1+VC1 = 6.26 ± 0.31% and B1+VC2 = 6.12 ± 0.02%), but only the first two of the mixed OM inputs were significantly different from the control soil (Table 4; p < 0.05). All other OM inputs led to a significant relative decrease compared to the control

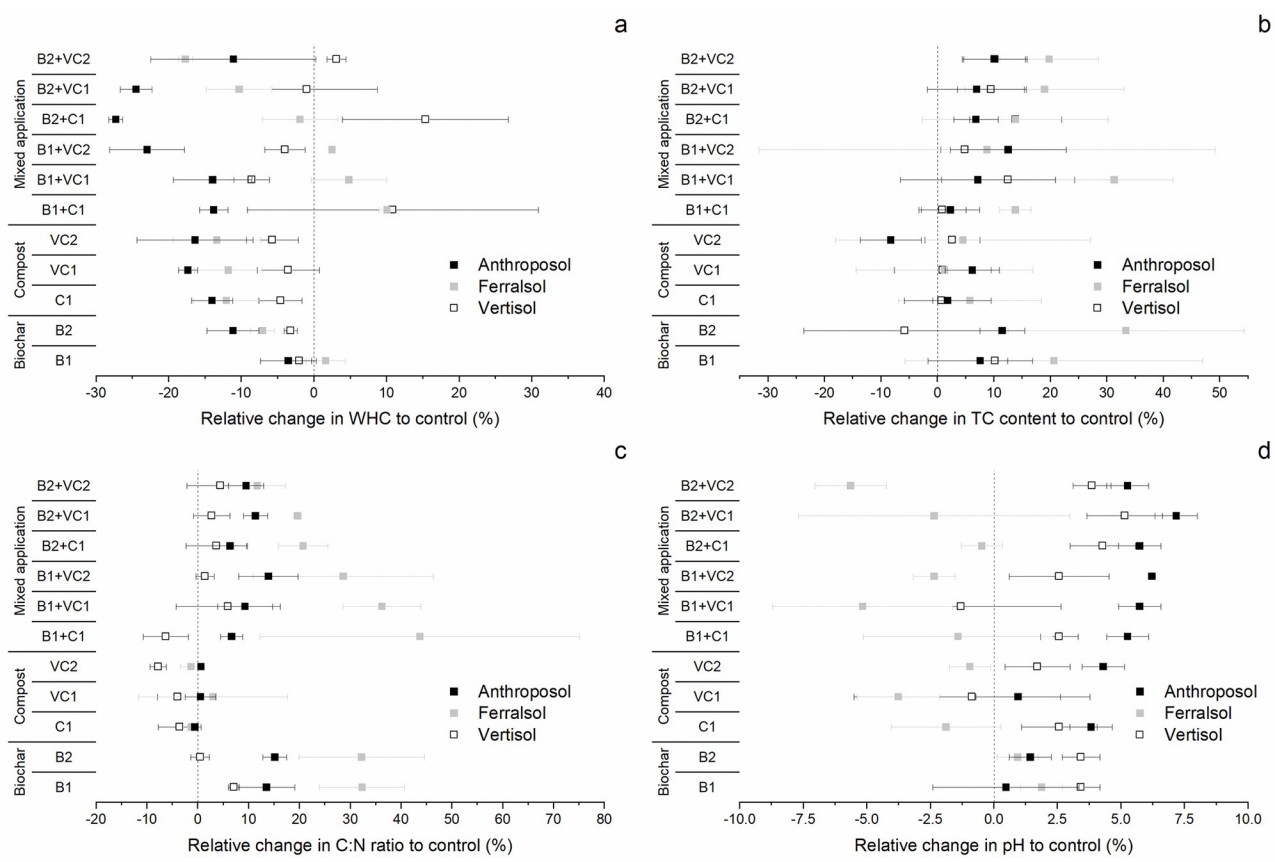

**Fig 1. Relative changes (%) of a) water-holding capacity (WHC), b) total carbon (TC) content, c) C:N ratio and d) pH after application of sole (biochar (B1: Coconut shell, B2: Rice husk) and composts (C1: Compost, VC1 and VC2: Vermicomposts)) and mixed inputs (combination of each biochar with one of the composts) to control Anthroposol, Ferralsol and Vertisol.** Values are means (n = 3) ± one standard deviation.

(-1.91 ± 5.20% to -17.70 ± 0.96%; Fig 1a), with lowest WHC when rice husk biochar was applied in combination with vermicompost to the Ferralsol (B2+VC2 = 4.91 ± 0.06%; p < 0.05). For the Vertisol, most OM inputs led to a slight, not significant decrease in WHC (initial WHC = 16.38 ± 1.20%; comparable to the volumetric water contents of Vertisols studied elsewhere in the research area [51]). This was not the case for three treatments where the WHC increased for B2+VC2 = 16.89 ± 0.21%, B1+C1 = 18.16 ± 3.28% and B2+-C1 = 18.90 ± 1.87%; however only significantly in the last case (Table 4; p < 0.05). Overall, highest increases were observed for coconut shell biochar + compost added to the Ferralsol (+10.07 ± 1.14%) and Vertisol (+10.87 ± 20.04%), and rice husk biochar and compost added to the Vertisol (+15.4 ± 11.42%), but with considerable variability (Fig 1a). The positive responses of soil WHC after a combined application of biochar with either compost [68] or vermicompost [69] were also found in previous research. Application of OM to soils is expected to improve soil moisture content, micro- and macro-pore density, and thus infiltration [70–72]. Despite its initial very low WHC and general poor characteristics (Table 1), the Ferralsol generally did not benefit from these OM applications. This contradicts several studies where biochar [73–75] or compost [69–72] increased water storage in soils, particularly also for sandy soils with low initial WHC common in tropical regions [38, 73, 74] and also for other Ferralsols [68]. In addition, we observed the strongest negative responses in soil WHC in the presence of rice husk biochar, which contradicts a study using the same biochar feedstock [75].

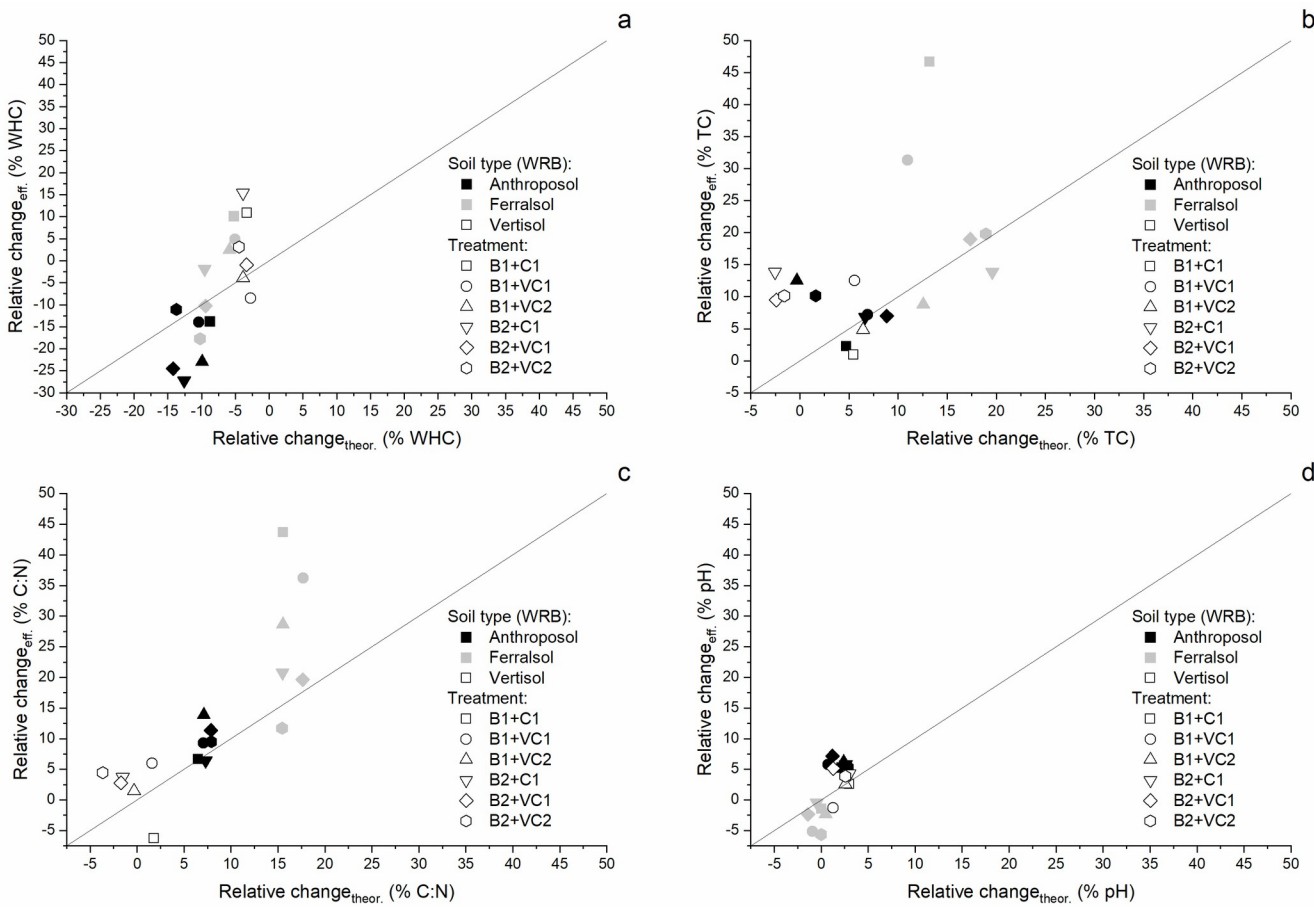

**Fig 2. Comparison of effectively measured relative changes (%) of mixed inputs (always a biochar mixed with a compost prior to application) with the theoretically expected weighted averaged changes (%) of sole inputs after application to control Anthroposol, Ferralsol and Vertisol for a) water-holding capacity (WHC), b) total carbon (TC) content, c) C:N ratio and d) pH using a similar approach as in [66].** Organic matter inputs include biochar (B1: coconut shell, B2: rice husk) and composts (C1: compost, VC1 and VC2: vermicomposts) treatments. Values are means (n = 3). Solid line represents 1:1-line.

Reduction of soil WHC in presence of biochar may be linked to initial hydrophobic properties of its surface, which disappear over the long-term with the oxidation of hydrophobic functional groups [73, 76]. Another possible explanation for the low and/or mostly negative WHC responses observed in our study could be related to the application rate of 10 t OM ha$^{-1}$ (based on local conditions of farmers in the research area), which is much lower compared to rates (133 and 40 t biochar ha$^{-1}$ respectively) in controlled-condition experiments [74] or summarized in a recent review [38].

**Table 3. P-values from the linear model output (one-way ANOVA) comparing the studied parameters (water-holding capacity (WHC; in %), total carbon content (TC; in %), C:N ratio and pH (unitless)) between the soils and treatments, and their interaction (n = 3, p.adj. = 0.05).** For details on the statistical approach, see Material and methods section.

|  | WHC [%] | TC [%] | C:N | pH [-] |
|---|---|---|---|---|
| **Soil** | $< 2.0 \times 10^{-16}$ | $< 2.0 \times 10^{-16}$ | $< 2.0 \times 10^{-16}$ | $< 2.0 \times 10^{-16}$ |
| **OM input** | 0.0006 | 0.0395 | $1.02 \times 10^{-9}$ | $6.79 \times 10^{-5}$ |
| **Soil: OM input** | $5.58 \times 10^{-6}$ | 0.2512 | 0.0016 | $1.35 \times 10^{-5}$ |

**Table 4. Descriptive statistics (means ± standard deviation (n = 3)) for the measured parameters (water-holding capacity (WHC; in %), total carbon content (TC; in %), C:N ratio and pH (unitless) for the control Anthroposol, Ferralsol and Vertisol and after treatment of these three soils with biochar (B1: Coconut shell, B2: Rice husk), composts (C1: Compost, VC1 and VC2: Vermicomposts) and combined OM inputs.** Letters (= group) indicate significant differences (p < 0.05) in the linear model output (one-way ANOVA: n = 3, p.adj = 0.05). For details on the statistical approach, see Material and methods section.

| | WHC [%] | | | TC [%] | | | C:N ratio | | | pH [-] | | |
|---|---|---|---|---|---|---|---|---|---|---|---|---|
| | mean | std | group | mean | std | group | mean | std | group | mean | std | group |
| **Anthroposol** | | | | | | | | | | | | |
| Control | 14.02 | 0.69 | a | 1.13 | 0.03 | ab | 9.55 | 0.10 | e | 6.97 | 0.06 | d |
| B1 | 13.52 | 0.54 | ab | 1.22 | 0.10 | a | 10.84 | 0.53 | ab | 7.00 | 0.20 | d |
| B2 | 12.46 | 0.50 | bc | 1.26 | 0.04 | a | 11.00 | 0.22 | a | 7.07 | 0.06 | d |
| C1 | 12.05 | 0.40 | c | 1.15 | 0.09 | ab | 9.50 | 0.03 | e | 7.23 | 0.06 | c |
| VC1 | 11.59 | 0.18 | cd | 1.20 | 0.05 | a | 9.60 | 0.28 | e | 7.03 | 0.11 | d |
| VC2 | 11.72 | 1.12 | cd | 1.04 | 0.06 | b | 9.61 | 0.03 | de | 7.27 | 0.06 | bc |
| B1+C1 | 12.08 | 0.27 | c | 1.16 | 0.06 | ab | 10.19 | 0.21 | c | 7.33 | 0.06 | abc |
| B1+VC1 | 12.06 | 0.76 | c | 1.21 | 0.15 | a | 10.44 | 0.51 | bc | 7.37 | 0.06 | abc |
| B1+VC2 | 10.80 | 0.72 | de | 1.27 | 0.11 | a | 10.88 | 0.56 | ab | 7.40 | 0.00 | ab |
| B2+C1 | 10.12 | 0.14 | e | 1.21 | 0.04 | a | 10.16 | 0.32 | cd | 7.37 | 0.06 | abc |
| B2+VC1 | 10.58 | 0.30 | de | 1.21 | 0.10 | a | 10.63 | 0.23 | abc | 7.46 | 0.06 | a |
| B2+VC2 | 12.46 | 1.59 | bc | 1.25 | 0.06 | a | 10.46 | 0.33 | abc | 7.33 | 0.06 | abc |
| **Ferralsol** | | | | | | | | | | | | |
| Control | 5.97 | 0.24 | bc | 0.45 | 0.03 | b | 8.64 | 0.37 | de | 7.10 | 0.10 | abc |
| B1 | 6.07 | 0.16 | bc | 0.54 | 0.12 | ab | 11.43 | 0.72 | abc | 7.23 | 0.06 | a |
| B2 | 5.55 | 0.10 | de | 0.60 | 0.09 | ab | 11.42 | 1.06 | abc | 7.17 | 0.06 | ab |
| C1 | 5.25 | 0.27 | ef | 0.47 | 0.06 | ab | 8.53 | 0.02 | e | 6.97 | 0.15 | abcde |
| VC1 | 5.27 | 0.28 | ef | 0.45 | 0.07 | b | 8.90 | 1.27 | de | 6.83 | 0.11 | cde |
| VC2 | 5.17 | 0.36 | ef | 0.47 | 0.10 | ab | 8.52 | 0.18 | e | 7.03 | 0.06 | abc |
| B1+C1 | 6.57 | 0.07 | a | 0.66 | 0.26 | a | 12.42 | 2.71 | a | 7.00 | 0.26 | de |
| B1+VC1 | 6.26 | 0.31 | ab | 0.59 | 0.05 | ab | 11.77 | 0.66 | ab | 6.73 | 0.25 | de |
| B1+VC2 | 6.12 | 0.02 | bc | 0.49 | 0.18 | ab | 11.11 | 1.53 | abc | 6.93 | 0.06 | bcde |
| B2+C1 | 5.86 | 0.31 | cd | 0.51 | 0.07 | ab | 10.43 | 0.42 | bcd | 7.07 | 0.06 | abc |
| B2+VC1 | 5.36 | 0.27 | e | 0.53 | 0.06 | ab | 10.34 | 0.07 | bcde | 6.93 | 0.38 | de |
| B2+VC2 | 4.91 | 0.06 | f | 0.54 | 0.04 | ab | 9.65 | 0.48 | cde | 6.70 | 0.10 | e |
| **Vertisol** | | | | | | | | | | | | |
| Control | 16.38 | 1.20 | bc | 1.56 | 0.05 | bc | 15.89 | 0.30 | abcd | 7.77 | 0.11 | bcd |
| B1 | 16.06 | 0.28 | bc | 1.72 | 0.03 | ab | 17.02 | 0.17 | a | 8.03 | 0.06 | ab |
| B2 | 15.86 | 0.15 | c | 1.47 | 0.28 | c | 15.97 | 0.29 | abc | 8.03 | 0.06 | ab |
| C1 | 15.63 | 0.48 | c | 1.57 | 0.02 | abc | 15.33 | 0.67 | bcd | 7.97 | 0.11 | abc |
| VC1 | 15.81 | 0.70 | c | 1.57 | 0.13 | abc | 15.27 | 0.64 | bcd | 7.70 | 0.36 | cd |
| VC2 | 15.45 | 0.58 | c | 1.60 | 0.07 | abc | 14.66 | 0.25 | d | 7.90 | 0.10 | abcd |
| B1+C1 | 18.16 | 3.28 | ab | 1.57 | 0.06 | abc | 14.90 | 0.71 | cd | 7.97 | 0.06 | abc |
| B1+VC1 | 14.98 | 0.40 | c | 1.75 | 0.18 | ab | 16.84 | 1.62 | a | 7.67 | 0.30 | d |
| B1+VC2 | 15.73 | 0.46 | c | 1.63 | 0.07 | abc | 16.12 | 0.28 | abc | 7.97 | 0.15 | abc |
| B2+C1 | 18.9 | 1.87 | a | 1.77 | 0.13 | a | 16.48 | 0.95 | ab | 8.10 | 0.10 | a |
| B2+VC1 | 16.23 | 1.59 | bc | 1.70 | 0.09 | ab | 16.34 | 0.57 | ab | 8.17 | 0.11 | a |
| B2+VC2 | 16.89 | 0.21 | abc | 1.71 | 0.08 | ab | 16.60 | 1.04 | a | 8.07 | 0.06 | a |

Highest increases in TC were observed after application of biochar alone or in combination with composts (Fig 1b and Table 4), as expected considering the high C content and relative stability of these substrates (B1 = 85.73 ± 4.65% and B2 = 42.66 ± 0.75%; Table 2) [37, 38]. Previous research has even shown a linear relationship between C added by OM including

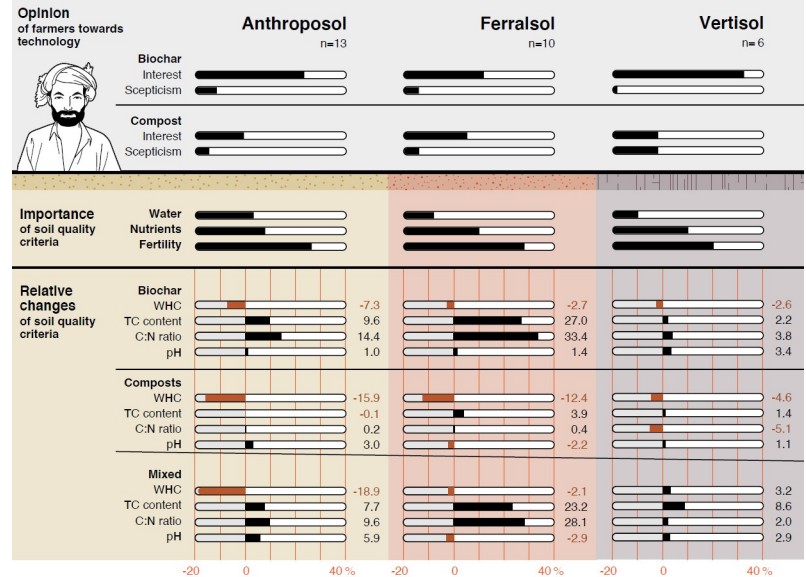

**Fig 3. Comparison of the total interest and skepticism of farmers cultivating Anthroposols, Ferralsols and Vertisols towards biochar and compost technologies, the importance of selected soil quality criteria (water, nutrients and fertility) rated by farmers and the relative changes (%) of the water-holding capacity (WHC), total carbon (TC) content, C:N ratio and pH after application of sole (biochar (B1: Coconut shell, B2: Rice husk) and composts (C1: Compost, VC1 and VC2: Vermicomposts)) and mixed inputs (combination of each biochar with one of the composts) to control Anthroposol, Ferralsol and Vertisol.** ® Illustration by Tara von Grebel, Information Technology, MELS/SIVIC, University of Zurich.

biochars and TC contents of the amended soil [37, 68]. Even if all OM inputs except for one of the vermicomposts (VC2 = 1.04 ± 0.06%; possibly due to fast decomposition of the added and/ or native C [40]) increased the C content of the control Anthroposol (1.13 ± 0.03%), these changes were not significant (Table 4; $p > 0.05$). Highest relative changes in TC contents (+12.54 ± 10.29%; Fig 1b) were observed after applying coconut shell biochar mixed with vermicompost for the Anthroposol (B1+VC2 = 1.27 ± 0.11%). Also in case of the Ferralsol with the lowest initial C content (0.45 ± 0.03%), all OM inputs, except for the combination of coconut shell biochar and compost (B1+C1 = 0.66 ± 0.26%; $p < 0.05$), resulted in no significant increase in the C content of the soil (Table 4). For the Vertisol (initial C content of 1.56 ± 0.05%), only the combination of rice husk biochar and compost (1.77 ± 0.13% (13.84 ± 8.15%; Fig 1b) led to a significant increase in C contents (Table 4; $p < 0.05$). For this specific parameter, the initial C contents of soil, lower for the Ferralsol, explain directly our observations that for a given OM input, the relative increases compared to control soils were mostly highest for the Ferralsol and lowest for the Vertisol (Fig 1b). The relative changes observed for the Ferralsol are in the same magnitude as for a Ferralsol amended with different mixtures of biochar, composts and fertilizer in tropical, northern Australia [68] or lateritic, low-quality soils in plot studies in Tamil Nadu, India [18]. However, the effects were not strictly additive, but rather synergetic for certain OM inputs and soils (Fig 2b) [37]. Considering the relative short duration of our experiment, it is probable that biochar protected the associated compost from decomposition, which is in line with our observation of higher soil C contents after 70 days of incubation for biochars mixed with composts compared to composts only [40]. The amplitude of the TC increase was also very dependent on the type of biochar and compost and the initial soil [38]. However, no clear trend can be associated to these changes and most positive responses in C contents of amended soils where not significant

compared to the three control soils (Anthroposol, Ferralsol and Vertisol), which is in line with similar incubation studies with biochar and tropical soils [26].

C:N ratios increased when biochar was applied alone or in mixture with composts, and generally remained close to the initial soil C:N value with no significant difference after application of composts alone (Figs 1c and 2c; Table 3). The Anthroposol with an initial C:N ratio of $9.55 \pm 0.10$ showed highest increases when biochar was applied alone (B1 = $10.84 \pm 0.53$ ($+13.54 \pm 5.56\%$) and B2 = $11.00 \pm 0.22$ ($+15.16 \pm 2.33\%$)) or in combination with vermicompost (B1+VC2 = $10.88 \pm 0.56$ ($+13.91 \pm 5.82\%$)). All OM inputs including biochar significantly increased the C:N ratio of the Anthroposol (Table 3; $p < 0.05$). For the Ferralsol, the same effect for the application of biochar alone or in combination with composts could be observed on the initial soil C:N ratio as for the Anthroposol (Fig 1c and Table 3). In contrast however, only biochar alone and coconut shell biochar with composts (up to $12.42 \pm 2.71$ ($+43.72 \pm 31.43\%$) for B1+C1) were significantly different from the control, but not the combination of rice husk biochar and composts (Table 3; at $p < 0.05$). In addition, the OM inputs had generally more influence on the C:N ratio of the Ferralsol (mostly > 20% increase; Figs 1c and 2c) than the Anthroposol. No differences between the Vertisol treated with OM and the control soil could be observed in our data in terms of C:N (Table 3; $p > 0.05$), probably because the initial C:N of the Vertisol was already high ($15.89 \pm 0.30$). The highest effect was achieved with application of coconut shell biochar (e.g. B1 = $17.02 \pm 0.17$). Our observations of steady C:N ratios after application of composts [39, 77], and increases after application of biochar alone [37, 78] or in combination with composts are in line with previous field and/or laboratory studies [77, 79]. Considering C:N values of the OM inputs (Table 2), we expected that compost would compensate for C:N increase due to biochar. On the contrary, the C:N became very high in presence of biochar, in particular for the Ferralsol (Fig 1c). Considering that this nitrogen can be a source of nutrients for plants on the long-term, the increase of C at the detriment of N may be a potential issue for soil fertility, as already pointed out in the literature [80, 81]. However, even if the C:N increases in presence of sole ore mixed biochar applications with composts, the (surface) properties of biochar may have positive effects on the N availability through reduced losses of ammonium and nitrate by volatilization or leaching [69, 79, 82].

All OM inputs resulted in a pH increase of the Anthroposol compared to the control ($6.97 \pm 0.06$), but the effect was mostly only significant when biochar was applied in combination with composts and highest for rice husk biochar mixed with vermicompost (B2+VC1 = $7.46 \pm 0.06$ ($+7.18 \pm 0.83\%$); Fig 1d and Table 4). The Ferralsol included in our study had a higher pH ($7.10 \pm 0.10$) compared to the values generally reported for Ferralsols in literature [67, 68]. Only applying biochar alone slightly increased the pH ($+1.88 \pm 0.81\%$ (coconut shell) resp. $+0.94 \pm 0.81\%$ (rice husk)), while all other inputs led to a relative decrease by up to $-5.63 \pm 1.41\%$ for biochar in combination with composts (B2+VC2 = $6.70 \pm 0.10$; Table 4; $p < 0.05$). This initial high pH could have resulted from application of alkaline fertilizers (ashes from slash-and-burn agriculture) by the farmer prior to our sampling. The addition of rice husk biochar and composts resulted in a significant relative increase of up to $+5.15 \pm 1.49\%$ in the Vertisol ($+0.4$ in pH for B2+VC1 ($8.17 \pm 0.11$)), whereas all other OM inputs were not different from the control soil ($7.77 \pm 0.11$; Table 4). The effect of biochar application depends largely on intrinsic soil properties such as initial soil pH and cation exchange capacity, but also on biochar quality (pH, exchangeable base cation concentration, $CaCO_3$ content and therefore acid ($H^+$) buffer capacity) and quantities applied [37, 83, 84]. Biochar application results mainly in positive, increasing effects on pH of non-amended soils with a larger effect on acidic soils [37, 38]. This is what we observed here, with stronger increases in the Anthroposol. We did not observe differences in soil pH responses related to the biochar quality (input material, production conditions); probably because the differences in pH between soils were

not so strong resp. important (no significant differences in soil pH between OM inputs including coconut shell resp. rice husk biochar). In addition, and in contrast to the slight and significant positive effect size of biochar application on soil pH found in a global meta-analysis [37], we could not observe any significant response in soil pH after biochar was applied alone (Table 3). Application of composts may lead to reduction of soil pH through the release of organic acids during decomposition of added OM or salts leaching [85], or an increase because of the production of base cations or hydroxide anions during decomposition. Our data showed effects in both directions: a decrease of the pH of the Ferralsol, and an increase of pH for Anthroposol and Vertisol (Fig 1b). Fig 2d shows that mixed application of biochar and composts resulted in a decreased pH of the Ferralsol, but increased pH for the Anthroposol and Vertisol. An increase in soil pH after application of mixtures of biochar and composts was also observed under field conditions for Acrisols in the tropics [39, 69] and in pot experiments [86, 87]. In contrast, and, in line with our findings, a Ferralsol in the tropics of northern Queensland also showed reduced or similar pH to control soils after mixed OM inputs [88], which may be the result of the increased mineralization of SOM after application of fresh OM to the Ferralsol with low TC content [89, 90].

Contrary to what was expected from the literature, and what was expected for the farmers, biochars alone or in mixture did not improve unilaterally the soil properties of these tropical soils, particularly of the Ferralsol, nicely highlighting the diverse responses of plant-soil systems to biochar applications [37, 38]. While it seems relatively simple to increase the TC content and to certain extends the pH of the soil [37], water and long-term nutrient status did not improve, or degraded with the OM inputs (Figs 1 and 2; Table 3). It is likely that the duration of the present incubation (70 days) could not capture the transition of biochar surfaces from hydrophobicity towards higher affinity to water [76], but increases in the WHC of biochar-amended soils on the long-term have already been proven in many cases [38, 73]; however, in terms of nutrients such as N, our observations are to be seen more critical if these are not plant-available on the longer-term [80, 81]. This divergent effect was even more marked for the Ferralsol, which was identified with lower fertility, so where the largest impact was expected [38, 91]. In addition, contrarily to our expectations, the mixture between composts and biochar did not compensate systematically the shortcomings of OM inputs when applied alone (as compared in Fig 2) [40]. While the TC content and C:N were almost systematically higher for the mix, showing a synergetic effect between the two inputs [79], this was soil-dependent for the WHC and for the pH. These two aspects highlight the versatility of biochar inputs to soil properties [37], and so the need to locally estimate its multi-seasonal impact, under specific pedo-climatic conditions [38, 42].

## Traditional (local) knowledge and practices of farmers in the Berambadi watershed regarding organic matter management

Tailor-made compost and biochar systems need to be adopted by the practices of rural farmer communities as well as their aspirations towards working with the new OMM techniques in their farming system [42, 43, 47]. To evaluate this, we conducted expert interviews (n = 14) with scientists, NGOs and private companies active in the field of agricultural development, and in-depth, qualitative interviews based on a standardized topic guide with farmers of the Berambadi watershed (n = 29), cultivating the same soil types used for the manipulative soil study (Anthroposol n = 13, Ferralsol n = 10 and Vertisol n = 6). The topic guide included questions about farming practices in general, about specific practices and perceptions regarding OMM (i.e. composting, slash-and-burn, other usage of OM) and finally on aspirations upon the introduction of new technologies like biochar (see Material and methods and S1 File).

Our empirical data from the interviews revealed that farmers used specific traditional practices of composting (97% of farmers). These could range from traditional techniques like Jeevamrutha (liquid compost made of ten liters of cow urine, ten kilograms of cow dung, two kilograms of black bean (powder), two kilograms of (black) Jaggery and one fist of soil, 17%) or farmyard manure preparation (97%) to more knowledge-intensive setups of composting (i.e. vermicomposting, 28%). Most of these practices have already been described in detail in the literature, also specific methods practiced in India, but they are normally highly variable locally [30]. Other traditional usage of OM included mulching (90%), forage for animals (52%), fuel for cooking (72%), application of river sediments (17%; see ref [53]) and the application of ash/charcoal from slash-and-burn agriculture or indoor cooking (59%), and these were also found in a farmer survey in Karnataka, India [33]. Farmers' knowledge about these OM inputs originated from traditions (family/community, practical experience, 72%), individuals (55%), local agricultural department (41%) or media (TV, books: 41%). Knowledge included awareness regarding the preparation (materials, setup), application (rate), timing and effects on soils (structure, moisture and nutrient contents, soil biota (earthworms)) and plants (growth dynamics and yield) (Quote 1).

> *Quote 1 (F4): «So when he introduces this into the soil, there's a rotation happening because of the worms and the soil. And then they form pores. Because of this, the soil becomes more nutrient and when it becomes more nutrient, it gives a better yield.»*

We further asked farmers about their knowledge on vermicomposting and biochar inputs. Regarding vermicomposting, we found a gap between farmers' knowledge and its practical application (Quote 2). Even if farmers had such knowledge (97%) and some of them were willing to apply the technology in their farming system (35%), the implementation level was found to be low according to our data (28%). Information on implementation levels provided by local scientists and NGO-workers in the study area (25% of farmers are using installed vermicomposting units) and in scientific literature [33] confirm our results. Farmers stressed their lack of practical experience (52%) or their negative experiences, i.e. weed growth, heat generation and short effect duration (21%) and lack of institutional guidance (14%; Table 5 and Quote 3).

> *Quote 2 (F18): «So he says, he makes a bed of the cow dung first and then he puts the green leaves. And then he puts the worms. And then he sprinkles it with water. Because it has to be cold. Else the worms die. And once this is done, these eat and then you have the compost.»*

> *Quote 3 (F22): «They have a communication problem from the officials. Because nobody guided him so that he can do this. And there is a problem in our administrative system and in the Karnataka government that whatever they want to do, is not reaching the farmers. So the implementation is not there. Nobody is there to guide them properly and communicate ideas to them.»*

According to our discussions with experts, biochar systems in India were limited to scientific studies and a broader public interest at the time of fieldwork, which is confirmed by recent reports on the topic [16, 92]. Interview data showed that only few farmers knew about biochar systems (31%) and even fewer who actively engaged in the concept (Quote 4). Nonetheless, even if the level of knowledge can be considered low, still many farmers showed great interest in biochar systems (83%, Fig 3) and 17% asked about implementing it in their farming system

**Table 5. Socio-economic (education, finance, technology) and agro-ecological (soil, plant, ecosystem) expectations and doubts of farmers in the Berambadi watershed upon the introduction of vermicompost and biochar technology in their farming systems.** Number in parentheses are percentage of farmers from all farmers (n = 29) who designated specific issues (%; VC = vermicompost, B = biochar).

| | | Expectation *(VC = 72%; B = 93%)* | Doubt *(VC = 48%; B = 52%)* |
|---|---|---|---|
| **Socio-economic** | Education | • Guidance, demonstration and long-term support *(VC = 41%; B = 48%)*<br>• Acquisition of practical knowledge/experience (preparation, application (rate), timing) *(VC = 21%; B = 55%)*<br>• Traditional practices as a basis (i.e. ash/charcoal for biochar applications *(B = 59%)*<br>• Easy and direct application *(VC = 3%; B = 45%)* | • Lack of institutional, long-term support *(VC = 14%; B = 7%)*<br>• Requirement of specific knowledge *(B = 7%)* |
| | Finance | • Purchase or free, institutional supply *(VC = 38%; B = 21%)*<br>• Investment capacity *(VC = 14%; B = 14%)*<br>• On-farm production *(VC = 7%; B = 21%)*<br>• Cost-efficiency and income generation *(VC = 10%)*<br>• Reduction of mineral fertilizer use *(B = 3%)* | • Resource availability (dependent on cultivated crops and tree availability) *(VC = 21%; B = 21%)*<br>• Extra cost and labor *(VC = 17%; B = 7%)*<br>• Resource competition (with other usages) *(VC = 7%; B = 7%)*<br>• Investment risk *(VC = 7%)* |
| | Technology | • Supply of tools (tank, pyrolysis unit, energy) *(VC = 14%; B = 17%)*<br>• Health improvements *(VC = 7%)* | • Duration/patience *(VC = 10%; B = 3%)* |
| **Agro-ecological** | Soil | • Soil fertility increase *(VC = 17%; B = 21%)* | • No need for new technology (soil microorganisms & traditional methods sufficient) *(VC = 10%; B = 7%)*<br>• Burned organic matter harms soil<br>  • Heat generation (only applicable with water) *(VC = 14%; B = 10%)*<br>  • Survival of earthworms/microorganisms *(VC = 10%; B = 7%)* |
| | Plant | • Sustain or increase crop growth/yield *(VC = 48%; B = 41%)*<br>• Weed control *(B = 3%)* | |
| | Ecosystem | • Suitable (and locally available) organic matter inputs *(B = 24%)*<br>• Non-hazardous for plant-soil system *(VC = 14%; B = 17%)*<br>• Reduction of fertilizer usage *(B = 3%)* | • Production and application quantity *(VC = 21%; B = 21%)*<br>• Usage of wood from on-farm trees or forests *(B = 14%)* |

(Quote 5) if guidance and support would be guaranteed (Table 5). Knowledge transfer through demonstration and long-term support are critical socio-economic barriers for successful implementation of technologies such as biochar systems [29, 48].

> *Quote 4 (F2): «Yes there is not a fertile. Just there is some media. Biochar is medium. It's not a fertile. Just media is that material is kept all the microorganisms to in holding capacity is there. So it will give the plants.»*

*Quote 5 (F15): «How come we have not heard about it and nobody has told us about it?»*

When discussing about biochar technology, farmers often related to their traditional (local) knowledge and practice of ash/charcoal application (59%; Table 5). Application of ash/charcoal was considered to be beneficial (28%) or hazardous (21%) for soil fertility and plant growth by farmers (Quotes 6 and 7), including positive (improvement in moisture and nutrient contents and soil structure) and negative (heat generation, death of earthworms, decreased crop growth) effects. The rationales of adopting slash-and-burn agriculture (or applying ash/charcoal to soils) of farmers in the research area are in line with the literature [93]. Furthermore, in accordance with the divided opinions of farmers towards the benefits and risks of this practice (Quotes 6 and 7), the sustainability of slash-and-burn agriculture is highly debated in the scientific literature until today [93–95].

*Quote 6 (F1): «Because of the water-holding capacity he is using the ash.»*

*Quote 7 (F18): «And the remains is an ash. Nobody puts that ash back into the soil because if you put that, you'll never get a yield. The plants will never grow well. So nobody uses that.»*

Farmers' perceptions about ash/charcoal application greatly influenced their attitude towards the introduction of biochar-based fertilizer systems, and farmers who engaged in these traditional practices were generally more open to biochar technology. The occurrence of such soil fertilization practices involving ash/charcoal applications from various sources (slash-burn agriculture, home cooking, and processing agricultural residues) could increase the implementation potential of biochar-based fertilization [43, 48]. A shift from slash-and-burn to slash-and-char farming practices could be viable since farmers possess significant knowledge about OMM and burning processes; however, site-specific concerns such as the availability and suitability of OM for charring and potential increased labor requirements due to collection of OM on fields instead of burning need to be addressed [12, 48].

## Farmers' aspirations towards the introduction of tailor-made biochar-based fertilizer systems

Interviewed experts pointed out three socio-economic and three agro-ecological categories that are crucial for a successful implementation of biochar-based fertilizer systems (Table 5). Socio-economic categories included educational issues (long-term guidance, knowledge requirements), financial issues (cost-efficiency, labor, level of effort, resource availability and competition) and technological issues (low-tech setups, duration). Agro-ecological categories included soil related issues (increase nutrient level and soil fertility), plant related issues (increase crop growth/yield) and ecosystem related issues (OM inputs, suitability for agro-ecosystems). Empirical data from interviews with farmers showed that farmers could precisely name their socio-economic and agro-ecological aspirations towards tailor-made OMM (Table 5, according to expert categories). Farmers especially emphasised their need for guidance and acquisition of practical knowledge in a participatory way when it came to new OMM (79%), and they expressed doubts about the long-term institutional support (21%), which constitute major adaptation barriers to technology [29, 48]. Another crucial socio-economic aspect concerns the implementation costs especially regarding additional labor requirements and investment capacities (48%) [42, 43]. Both farmers and experts stressed out resource availability and suitability as well as the competition with other local usages of OM as major constraints for tailor-made, site-specific biochar systems (Table 5), which was also emphasized in a recent review on biochar systems [38, 48]. Whether OM was perceived as a viable agronomic

input for OMM, including biochar production, depended on the site-specific cropping systems (farming system (organic or conventional), traditional OMM practices, type of plants grown) of individual farmers (Quotes 8 and 9) [17, 23]. The production of biochar from on-farm (horticulture) or forest trees was considered to be problematic and concerns were expressed regarding the need for collecting wood biomass when engaging with biochar technology [3]. In this regard, it could well be that implementation of biochar systems could compete with traditional practices of OM usage (as defined above) for resources [6], and needs to be considered in further agricultural development projects [48].

> *Quote 8 (F7): «Yes it is good that what we can put into the soil, the remains of the plants. He puts it into the soil.»*

> *Quote 9 (F23): «So he feels that if you remove that it's just a waste. So what can you do with it? So they burn it and put it into the field.»*

Interview data further revealed that improved soil fertility and crop growth/yield were the major agro-ecological aspirations of farmers towards vermicomposting and biochar technology (Table 5). These aspirations could be met with biochar-based fertilizers (see Figs 1 and 2 and explanations above) and literature suggests that the site-specific environmental (water scarcity if no bore well irrigation) and soil quality problems (low SOC, generally low pH, poor structure) of sub-humid tropical regions like Berambadi watershed could be tackled through OM inputs such as biochar [12, 38, 42].

## Matching the effect of biochar-based fertilizers on soil quality with farmers' aspirations on organic matter management

Interviews with farmers (Table 5) and the manipulative soil study (Figs 1 and 2) revealed that the site-specific contexts, to which OMM techniques such as biochar systems are intended to be applied to, are key for successful implementation and sustainable agricultural development as suggested also in recent literature [29, 38, 42]. Our dual interdisciplinary approach highlighted a series of strong clear trends as well as contradictions or mismatches between farmer's expectations and measured soil quality changes (Fig 3). The data from our socio-economic study indicated that farmers were generally more interested in biochar technology than composting irrespective of the farming system, which follows partially our soil quality evaluation (Fig 3). However, farmers cultivating more fertile Vertisols and/or farmers who experienced a stronger economic position based on bore well irrigation capacities (year round crop production; see ref [50]) and land ownership, showed a greater interest towards new technology like biochar, but a lower interest in existing, traditional (local) practices such as composting. Farmers who cultivated Ferralsols with poorer soil quality felt more confident with traditional (local) practices rather than biochar technology, and greater skepticism towards biochar indicated their low investment capacity and readiness to assume risks [29, 48]. According to these findings, there is a clear danger that new technology implementation will mainly benefit to the wealthiest, and may increase the gap between farmers within a community if innovations are not widely accessible to all [3].

Besides their interest and skepticism, farmers could further exactly rate the relative importance of soil quality criteria such as water and nutrient levels, and soil fertility in general (Fig 3), which allowed to assess which technology would match their aspirations (Table 5). Even if water represented a major problem in the study area [50, 52], farmers did not designate it as the major criteria during fieldwork (<50% mentioned water as a criteria), partially because irrigation is still possible year round using groundwater for farmers possessing bore wells

(compared to rain-fed irrigation) [50]; however, with depletion of groundwater resources and climate change, water availability will likely become a major concern for many more farmers in the semi-arid areas of southwestern India [52]. Biochar-based fertilizer (mixture of biochar and composts) only improve WHC for the Vertisol and Ferralsol in some cases, but not for the Anthroposol, which represented the soil cultivated by farmers that rated water as most important (Fig 3), but these effects likely change over longer-term [76]. More generally, while aspirations and global effects of biochar and compost are in line, we observed several mismatches at local level between the absolute soil quality criteria, the expectations from the farmers given a particular soil, and the potential of new OM inputs to improve the soil quality. These very local contradictions could only be detected by combining interviews and experimentations, so we can only strongly encourage such approaches to design the implementation of new cropping systems [38, 42].

## Conclusions

Our study demonstrates that OM inputs such as biochar for sustainable agricultural development need to be tailor-made for site-specific socio-economic and agro-ecological contexts. By applying an interdisciplinary research approach, we could identify OMM techniques that can increase the ecosystem services of soils with specific ecological challenges that needed to be solved in order to sustain or increase agricultural production and rural farmer livelihoods, and that simultaneously address the site-specific adaptation barriers and aspirations of farmer communities towards biochar technology.

Biochar-based fertilizers that include a mixture of biochar and other OM such as composts represented a viable solution for soil quality improvements of three tropical, agricultural soils (Anthroposol, Ferralsol and Vertisol) along with their suitability to incorporate traditional (local) knowledge and practices of small-scale farmers in a sub-humid tropical watershed. Including traditional (local) knowledge and practices related to OMM, farmers' aspirations towards new technologies and concerns regarding resource availability, suitability and competition for biochar-based fertilizers will ultimately increase the implementation potential of such applications for sustainable agricultural development.

The substantial interest of farmers in biochar technology, irrespective of farming systems, provides scope for in-depth field trials, where biochar-based fertilizers can directly be tested together with farmers, including the site-specific socio-economic and agro-ecological challenges faced by rural farmer communities in different environmental settings. Future research could therefore provide answers to questions regarding type, quality and rates of OM applied, the influence of a changing climate on tropical agro-ecosystems as well as future cropping strategies of rural farmer communities.

## Supporting information

**S1 File. Supplementary information for: Tailor-made biochar systems: Interdisciplinary evaluations of ecosystem services and farmer livelihoods in tropical agro-ecosystems.**
(DOCX)

## Acknowledgments

The authors acknowledge the assistance and translation of interviews during fieldwork of Rajani Chennu and the scientific assistance of the Indo-French Cell for Water Sciences, Indian Institute of Science, Bangalore. The authors further acknowledge the provision of OM inputs by Prof. Prakash Nagabovanalli (UAS Bangalore) and Karthik Vermicompost and Earthworm

Consultancy. Furthermore, SB and SA acknowledge the help of technical staff at the University of Zurich, Soil Science & Biogeochemistry group and at the Soil and Terrestrial Environmental Physics Lab at the Swiss Federal Institute of Technology (ETH) for assistance during laboratory work. Finally, SB and SA highly appreciate valuable comments on the final draft of this manuscript from Julia Le Noe, Núria Catalán and Marcus Schiedung.

## Author Contributions

**Conceptualization:** Severin-Luca Bellè, Samuel Abiven.

**Data curation:** Severin-Luca Bellè, Samuel Abiven.

**Formal analysis:** Severin-Luca Bellè, Norman Backhaus, Samuel Abiven.

**Funding acquisition:** Samuel Abiven.

**Investigation:** Severin-Luca Bellè.

**Methodology:** Severin-Luca Bellè, Samuel Abiven.

**Project administration:** Samuel Abiven.

**Resources:** Jean Riotte, Muddu Sekhar, Pascal Jouquet, Samuel Abiven.

**Supervision:** Norman Backhaus, Samuel Abiven.

**Validation:** Severin-Luca Bellè, Jean Riotte, Norman Backhaus, Muddu Sekhar, Pascal Jouquet, Samuel Abiven.

**Visualization:** Severin-Luca Bellè.

**Writing – original draft:** Severin-Luca Bellè.

**Writing – review & editing:** Severin-Luca Bellè, Jean Riotte, Norman Backhaus, Muddu Sekhar, Pascal Jouquet, Samuel Abiven.

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
