## [Decision Letter · Decision Letter 0]

29 Nov 2021

PONE-D-21-24354Tailor-made biochar systems: Interdisciplinary evaluations of ecosystem services and farmer livelihoods in tropical agro-ecosystemsPLOS ONE

Dear Dr. Abiven,

Thank you for submitting your manuscript to PLOS ONE. After careful consideration, we feel that it has merit but does not fully meet PLOS ONE’s publication criteria as it currently stands. Therefore, we invite you to submit a revised version of the manuscript that addresses the points raised during the review process.

We look forward to receiving your revised manuscript.

Kind regards,

Primo Proietti

Academic Editor

PLOS ONE

Journal Requirements:

2. Please provide additional details regarding participant consent. In the Methods section, please ensure that you have specified (1) whether consent was informed and (2) what type you obtained (for instance, written or verbal). If your study included minors, state whether you obtained consent from parents or guardians. If the need for consent was waived by the ethics committee, please include this information.

Additional Editor Comments:

Hello,

the manuscript is potentially interesting, but presents strong criticalities/deficiencies both in the general structure and in the presentation/discussion of the results. Authors should carefully consider reviewers' comments to make the work publishable.

All the best

Reviewers' comments:

Reviewer's Responses to Questions

**Comments to the Author**

1. Is the manuscript technically sound, and do the data support the conclusions?

Reviewer #1: Partly

Reviewer #2: Yes

2. Has the statistical analysis been performed appropriately and rigorously? 

Reviewer #1: N/A

Reviewer #2: Yes

3. Have the authors made all data underlying the findings in their manuscript fully available?

Reviewer #1: No

Reviewer #2: Yes

4. Is the manuscript presented in an intelligible fashion and written in standard English?

Reviewer #1: No

Reviewer #2: Yes

5. Review Comments to the Author

Reviewer #1: The paper presents a comparison of different soil amendments to increase its content of organic matter in an Indian region. The topic is of high interest and relevance because comparative tests of biochar with other organic fertilisers areof paramount importance. Nevertheless the presentation of the paper is scarce and it is recommended to check further the compliance with the authors guidelines. The following changes are adviced in order to reach the required quality for the paper to be published:

- condense the abstract and insert more quantitative data;

- insert a nomenclature reporting the symbols used in the equations and their units of measure and also the abbreviations used throughout the paper

- insert progressive numbers in the paragraphs of the manuscript

- avoid use lumped citations, see: [3,15,30,36,38], each citation has to be descirbed with a sentence;

- explain at the end of the introduction the novelty and the impact of the paper for the scientific community and the industrial sector;

- at the end of the materials and methods section insert please a paragraph on the analytical methods, in which you indicate: the type of analysis, the norms consulted, the instrument used (model, producer and origin);

- figures 1,2 and 3 are not provided, insert them in the text, also increase the amount of data shown;

- all the data shown in tables and figures, when experimentally derived have to show the standard deviation

- the statistical treatment of the data is not clear andd poorly debated in the materials and methods section

- before the conclucions insert a discussion section in which you compare the results of your study with those obtained in the literature.

- the definition and quality of the figures proposed in the supplementary materials are too low

- please evaluate to insert the data shown in the supplementary materials in the main manuscript. I think there is not need of the supplementary materials section, because the amount of results shown is already quite poor.

Reviewer #2: Line no. Page no. Remarks

64-69 3 Statement: Within climate zones, agriculture in tropical regions is highly relevant regarding food security at global scale, while being more vulnerable to global changes. The state of Karnataka, mostly located in sub-humid to semi-arid climates in the southern part of the Indian peninsula, is among the most vulnerable Indian states to climate change, with major parts prone to drought and soil degradation, and from these perspectives correspond to what is observed in many places in the tropics.

Remarks: Please make this sentence more precise with the prospective of global readers and divided it into smaller simple sentences.

69-70 3 Statement: Its irrigation system and hence agricultural production depends on a large extent on monsoonal rainfall patterns and groundwater irrigation.

Remarks: Please remove the highlighted words and reframe the sentence in grammatically correct manner.

87-90 4 Statement: Among existing OMM techniques, composting, vermicomposting (utilizing earthworms to digest pre-composted OM) and biochar have been proposed to increase soil fertility and agricultural productivity, while simultaneously reducing the environmental impact of agriculture and socio-economic dependency of farmers.

Remarks:

• Please reframe this statement into two statements, i.e., for composting and vermicomposting in one statement and biochar in the second statement.

• Also mention the process of production is degradation of lignocellulosic biomass in the first statement while the process of production is carbonization with pyrolysis in next statement.

95-98 5 Statement: In this case, inputs with synergic and antagonistic properties are added together to the soil, for example biochar and compost, combining amendment (physico-chemical capacity of biochar) and fertilizers (compost).

Remarks: Please cite appropriate reference for statement

149-156 7 Statement: Soil samples were taken in representative fields of interviewed farmers of three soil types to a depth of 20 cm (agricultural horizon). A random selection of sample spots were chosen and samples have then thoroughly been mixed to one composite sample for each soil type, air-dried and kept in sampling bags before transport and analysis. Sampling of the Ferralsol took place on a hillslope, whereas the Vertisol was taken directly from a field next to the riverbank (both in Gopalpura town). The Anthroposol represents a native Ferralsol where river (tank) sediments were applied [47] and was sampled in Berambadi town near the forest to the state border of Kerala.

Remarks: Please make this paragraph more precise and up to the mark with only relevant numbers and information.

199-200 9 Statement: The samples were preliminary stirred for 30 minutes and then pH was measured in the soil suspension.

Remarks: Please do mention the rate of stirring in rpm in 30 minutes

315-317 14 Statement: We did not observe differences related to the biochar quality, probably because the differences between soils were not so important.

Remarks: Please try to avoid such generalised statements or reframe it.

351-354 15 Statement: Our empirical data from the interviews revealed that farmers used specific traditional practices like composting (97 % of farmers), ranging from traditional techniques like Jeevamrutha (liquid compost, 17 %) or farmyard manure preparation (97 %) to more knowledge-intensive setups of composting (i.e. vermicomposting, 28 %).

Remark: Please mention the formulation or composition of Jeevamrutha used here. As this formulation changes a lot with the traditional knowledge of local people and local availability of resources.

492-495 23 Statement: Future research could therefore provide answers to questions regarding type, quality and rates of OM applied, the influence of a changing climate on tropical agro-ecosystems as well as future coping strategies of rural farmer communities.

Remarks: Please replace the word ‘coping’ with ‘cropping’

Remarks: Please cite the below mentioned papers in Introduction part as recent literature review on biochar and agricultural biotechnology based new interventions.

1. Maddalwar, S., Kumar Nayak, K., Kumar, M., & Singh, L. (2021). Plant microbial fuel cell: Opportunities, challenges, and prospects. Bioresource Technology, 341, 125772. https://doi.org/10.1016/j.biortech.2021.125772

2. Kumar, Manish, Shanta Dutta, Siming You, Gang Luo, Shicheng Zhang, Pau Loke Show, Ankush D. Sawarkar, Lal Singh, and Daniel CW Tsang. "A critical review on biochar for enhancing biogas production from anaerobic digestion of food waste and sludge." Journal of Cleaner Production (2021): 127143.

6. PLOS authors have the option to publish the peer review history of their article (what does this mean?). If published, this will include your full peer review and any attached files.

Reviewer #1: No

Reviewer #2: **Yes: **Dr. Lal Singh, Senior Scientist, CSIR-NEERI, Nagpur

---

## [Author Response · Author response to Decision Letter 0]

17 Dec 2021

Response to the editors and reviewers comments for the manuscript PONE-D-21-24354

Dear Prof. Dr. Proietti,

we thank you and the reviewer for your comments on our manuscript and for the opportunity to improve its quality. Please find all our responses to the comments below (in blue) according to the journal requirements.

Journal Requirements:

We have checked the PLOS ONE’s style requirements again carefully, and made adjustments were necessary. We hope that the formatting is now according to PLOS ONE’s requirements. 

2. Please provide additional details regarding participant consent. In the Methods section, please ensure that you have specified (1) whether consent was informed and (2) what type you obtained (for instance, written or verbal). If your study included minors, state whether you obtained consent from parents or guardians. If the need for consent was waived by the ethics committee, please include this information.

We appreciate this reminder. We have already stated the details regarding participants in the last paragraph of the Methods sections, where we state the Ethics guidelines that we followed in our study and about how participants were informed and asked for consent. Still, we have specified and rephrased this section accordingly to make it more precise and clear (see lines 282 to 290 in the revised manuscript). We have also added a reference on the “data protection in qualitative research”, which we followed during fieldwork and interviews with farmers (see lines 285-286 in the revised manuscript).

We thank for this reminder and we have provided the (reserved) DOI for our dataset on Zenodo in the cover letter. The data associated with this manuscript on Zenodo can then simply be transferred from a reserved DOI to a published DOI so that everyone can see and access it after the manuscript is accepted (open access). 

Data availability statement:

The data associated to this article is available online on Zenodo (10.5281/zenodo.5769577) and upon request from the corresponding author. Information on sensitive data from expert and farmer interviews (transcripts, codes, etc.) can be requested from the corresponding author. 

Comments from the Editors and Reviewers:

Additional Editor Comments:

Hello,

the manuscript is potentially interesting, but presents strong criticalities/deficiencies both in the general structure and in the presentation/discussion of the results. Authors should carefully consider reviewers' comments to make the work publishable.

All the best

We thank the editor for considering our manuscript for publication in PLOS ONE and for the opportunity to resubmit our work after revision. We have carefully addressed all reviewers’ comments (see replies to comments below and revisions in the manuscript). We have especially improved the structure of the manuscript and worked on many aspects of the introduction and results & discussion as suggested by the reviewers. 

Reviewer #1: 

The paper presents a comparison of different soil amendments to increase its content of organic matter in an Indian region. The topic is of high interest and relevance because comparative tests of biochar with other organic fertilisers are of paramount importance. Nevertheless the presentation of the paper is scarce and it is recommended to check further the compliance with the authors guidelines. 

We thank the reviewer for the in-depth, constructive feedbacks on our manuscript and the propositions made below. We have carefully addressed all mentioned revisions below and tried to improve the presentation of the paper where necessary, especially regarding the amount of results and discussion presented. We have also carefully checked that our manuscript is in accordance with the authors guidelines of PLOS ONE and we modified the manuscript accordingly. 

The following changes are adviced in order to reach the required quality for the paper to be published:

We thank the reviewer again for all the in-depth comments on our manuscript. We have replied to each of them separately (see below):

- condense the abstract and insert more quantitative data;

We agree with the reviewer and carefully reworked the abstract by including more quantitative results as well as by rephrasing and condensing it to the most important information. See lines 24 to 46 in the revised manuscript.

- insert a nomenclature reporting the symbols used in the equations and their units of measure and also the abbreviations used throughout the paper

We thank the reviewer for this comment. We added the definition of the terms used in the equation 1 but, we believe that a nomenclature is not needed because all abbreviations are defined and explained upon first mentioning and we tried to reduce them to a minimum. In regards to the equation, we have updated the equation and the corresponding text so that the units of measure are directly introduced and defined at the appropriate place (see lines 195 to 209 in the revised manuscript). We believe that with these changes, it should be enough information for the reader. We actually have rarely seen a nomenclature in PLOS ONE if only few abbreviations are used. 

- insert progressive numbers in the paragraphs of the manuscript

We thank the reviewer for this comment. As far as we were able to check in the guidelines, PLOS ONE does not include progressive numbers of sections or paragraphs, and numbers rarely appear in PLOS ONE articles.

- avoid use lumped citations, see: [3,15,30,36,38], each citation has to be descirbed with a sentence;

We agree with the reviewers comment and carefully revised the references, especially those with lumped citations. See for example lines 52, 56, 61, 66, 70, 79, 81, 89-90, 112, 115, 122, 148, 439, 473. 

Since we have substantially worked on the results and discussion section (also adding more discussion points and literature), references were updated accordingly, but are not listed above. 

In addition, we have also updated our manuscript with some more recent publications about the topic as well as suggested references by reviewer 2. 

- explain at the end of the introduction the novelty and the impact of the paper for the scientific community and the industrial sector;

We thank the reviewer for this constructive suggestion. We have added a couple of sentences explaining the novelty and impact of our manuscript for the scientific community and industry in the last paragraph of the introduction starting from lines 114 to 129 in the revised manuscript. In doing so, we have also rephrased and restructured the paragraph to have a better reading flow. 

“The challenges described above call for an agriculture that fulfills in the same time ecological, socio-economic as well as political aspects of sustainability [11,44]. The benefits of OMM according to soil types and agro-ecosystems as well as the site-specific, socio-economic barriers are currently studied separately, in particular regarding applicability and alignment with the agricultural sector [29,45]. However, there is an urgent need to connect these evaluations in an interdisciplinary effort across multiple scientific communities from the natural and social sciences in order to develop and implement innovative cropping systems adaptive to specific agro-ecosystems that improve rural farmers’ livelihoods in parallel to soil ecosystem services [46–48]. We propose a novel, joint assessment of soil quality and farmers’ aspiration in order to evaluate the conformity of OMM to existing rural farming communities by using a variety of complementary methodologies rooted in human geography and the soil sciences. Our interdisciplinary approach could be seen as a first practical example on how to link agro-ecological and socio-economic questions in agricultural research that can form the basis for future in-depth, long-term field trials on OMM together with local farmer communities in tropical regions [46,47]. This could ultimately help to develop sustainable agricultural technologies for the development or local agricultural and industrial sectors.”

- at the end of the materials and methods section insert please a paragraph on the analytical methods, in which you indicate: the type of analysis, the norms consulted, the instrument used (model, producer and origin);

In the section “soil analysis”, all types of analysis and instrumentation are reported and described. We have specified these more clearly and added relevant information on the instruments (model, producer and origin) where applicable and necessary. We have also added information on the norms used for the individual methods (or additional information) if applicable. See the changes made in lines 218 to 236 in the revised manuscript. 

- figures 1,2 and 3 are not provided, insert them in the text, also increase the amount of data shown;

According to our knowledge and the authors’ guidelines, figures should be provided as individual, separate files in the online submission system and not be inserted in the main text body. We have placed the figure caption as placeholders in the main text of the body, but be believe that the figures are also accessible during the reviewing process. See the entry on figures in the guidelines of PLOS ONE:

“Do not include figures in the main manuscript file. Each figure must be prepared and submitted as an individual file. Cite figures in ascending numeric order at first appearance in the manuscript file.”

We agree with the reviewer that more (quantitative) data can be included in the main body of the text. We have added relevant data in the abstract, results& discussion section where appropriate, and also included the two tables with the measured parameters and statistical output from the supplementary material in the main body of the manuscript (see lines 299-303 on page 13-14 and lines 373-379 on pages 16-18 in the revised manuscript). 

- all the data shown in tables and figures, when experimentally derived have to show the standard deviation

We thank the reviewer for this comment. The standard deviation of the mean is provided for the Tables 1,2,3 and 4, and Figure 1 where quantitative data of the soils, organic matter and the incubation results are presented, all derived from the fieldwork and the incubation experiment. Table 5 and Figure 3 represent qualitative information from the farmers interview and Figure 2 and 3 somewhat qualitative information (relative changes etc.) from the incubation experiment. Thus, we refrain from reporting statistical information on qualitative data/information, especially for the qualitative data of the farmers interview. In this regard, see for example the ongoing debate on quantification of qualitative data and about mixed method approaches in qualitative research, e.g.

Maxwell JA. Using Numbers in Qualitative Research. Qual Inq. 2010;16: 475–482. doi:10.1177/1077800410364740

While it would be possible to use statistics on the qualitative data, we refrain from doing so because the numbers are based on interview data and codes, which do not represent measured experimental data. 

Nonetheless, we agree with the reviewer (see also next comment by reviewer 1) that the statistical analysis or the absence of it in terms of qualitative data needs to be clearly explained and debated accordingly. We therefore added an explanation to the section on “statistical analysis” in lines 249 to 254 in the revised manuscript. 

In addition, since we added more quantitative data in the results & discussion section, we also included the standard deviation along the mean for values reported in the main text. 

- the statistical treatment of the data is not clear andd poorly debated in the materials and methods section

We thank the reviewer for this comment and we have worked on the section on “statistical analysis”. We have rephrased and explained the statistical approach in more detail and added an explanation and reflection on the statistics respective absence of statistics for the qualitative data accordingly in lines 238 to 254 in the revised manuscript. 

“We performed an analysis of variance for the full dataset derived from the incubation experiment, testing for significant differences of measured soil properties (WHC, TC, C:N, pH) between soils, OM and their interactions (two-way ANOVA, n = 3). Subsequently, we performed another analysis of variance on the measured soil properties (WHC, TC, C:N, pH) for each soil type separately, with the OM inputs as the independent variable (one-way ANOVA, n = 3). For both ANOVA’s, we used Levene’s test to check the assumption of homogeneity of variance (center = mean), a Shapiro-Wilk test on the ANOVA residuals to check for the assumption of normality and a Fisher’s least significant difference (LSD) post-hoc test to check for significance (alpha = 0.05, p.adj. = bonferroni). Statistical analysis was carried out using R Studio 1.3.1093 (2009-2020), R Version 4.0.3 (R Core Team, 2020). All values reported represent means with one standard deviation. 

While it would have been possible to use simple, descriptive statistics on the numbers derived from the coding of empirical data of farmers’ interviews, we refrained from doing so because these numbers (mainly percentages) represent assigned codes to text sequences that originate from the statements of farmers during the interviews, and not measured data (see below). However, there is an ongoing debate in qualitative research if simple statistics can be used for qualitative data [55].”

- before the conclucions insert a discussion section in which you compare the results of your study with those obtained in the literature.

We thank the reviewer for this suggestion and we agree that the results should be more discussed in the light of literature. We have increased and improved the amount of comparison between our results and the literature in the result and discussion sections where appropriate, but we do believe that with this there is no further need of a separate section of comparison at the end of the results & discussion. However, we have improved the last paragraph in the results and discussion section before the conclusions and added relevant comparisons with the literature there. Additional discussion points were added at several places, see for example lines 293-298, 323-325, 328-332, 335-338, 346-347, 357-363, 366-367, 369-372, 396-399, 404-407, 425-431, 437-441, 452-453, 455-460, 486-488, 490-491, 529-534, 544-548, 557-561 and some minor changes made in the results & discussion in the revised manuscript. 

- the definition and quality of the figures proposed in the supplementary materials are too low

There are no figures in the supplementary material. We therefore cannot answer this review comment accordingly. Nonetheless, we have moved the two tables with the measured parameters and statistical information, as well as the farmers quotes, from the supplementary material to the main text body and in doing so adjusted/updated the quality and format of the tables accordingly. In addition, we have rephrased the definition of the supplementary material (topic guides of interviews) more precisely if this is what the reviewer was referring to.

- please evaluate to insert the data shown in the supplementary materials in the main manuscript. I think there is not need of the supplementary materials section, because the amount of results shown is already quite poor.

We thank the reviewer for this final input and we agree that more data can be added to the result section of the manuscript. Thus, we have moved the two tables from the supplementary material to the main body of the manuscript and finally included relevant data in the text were appropriate to increase the amount of results shown as suggested by the reviewer. We have also moved the nine quotes from the farmer interviews to the main body of the text, since these also represent important findings or highlight the results of the interview. 

Nonetheless, we believe that the topic guide of interviews should remain in the supplementary material since these are quite long information, mainly relevant to understand the approach, methodology and results of the qualitative part of the manuscript. 

Reviewer #2: Line no. Page no. Remarks

64-69 3 Statement: Within climate zones, agriculture in tropical regions is highly relevant regarding food security at global scale, while being more vulnerable to global changes. The state of Karnataka, mostly located in sub-humid to semi-arid climates in the southern part of the Indian peninsula, is among the most vulnerable Indian states to climate change, with major parts prone to drought and soil degradation, and from these perspectives correspond to what is observed in many places in the tropics. 

Remarks: Please make this sentence more precise with the prospective of global readers and divided it into smaller simple sentences.

We thank the reviewer for this comment. We have rephrased the text into shorter sentences so that it is easier to read and is more precise in the light of the reviewers comment. We have also tried to rephrase the sentences in a more “global perspective” or that the content of the sentences relate to a broader, more global approach. See lines 67 to 73 in the revised manuscript.

69-70 3 Statement: Its irrigation system and hence agricultural production depends on a large extent on monsoonal rainfall patterns and groundwater irrigation. 

Remarks: Please remove the highlighted words and reframe the sentence in grammatically correct manner.

We thank the reviewer for this comment. We have rephrased the sentence to make it more clear and concise, see lines 73 to 75 in the revised manuscript. 

87-90 4 Statement: Among existing OMM techniques, composting, vermicomposting (utilizing earthworms to digest pre-composted OM) and biochar have been proposed to increase soil fertility and agricultural productivity, while simultaneously reducing the environmental impact of agriculture and socio-economic dependency of farmers.

Remarks:

• Please reframe this statement into two statements, i.e., for composting and vermicomposting in one statement and biochar in the second statement.

• Also mention the process of production is degradation of lignocellulosic biomass in the first statement while the process of production is carbonization with pyrolysis in next statement.

We thank the reviewer for these two suggestions and we have now split the sentence into two containing the information on composting/vermicomposting in the first sentence, and on biochar in the second sentence. We have further included the information mentioned in point two in the respective sentences and added relevant references for the production for each composting/vermicomposting and biochar respectively. After these changes, we have further restructured the paragraph accordingly. See lines 91 to 104 in the revised manuscript. 

95-98 5 Statement: In this case, inputs with synergic and antagonistic properties are added together to the soil, for example biochar and compost, combining amendment (physico-chemical capacity of biochar) and fertilizers (compost).

Remarks: Please cite appropriate reference for statement

We thank the reviewer for this helpful comment and have added an appropriate reference. In fact, both sentences in lines 100-102 and 102-104 refer to the reference 41, so we moved it to the end of the paragraph and added a second, most recent reference (reference 38):

Ref. 38: Joseph S, Cowie AL, Van Zwieten L, Bolan N, Budai A, Buss W, et al. How biochar works, and when it doesn’t: A review of mechanisms controlling soil and plant responses to biochar. GCB Bioenergy. 2021; gcbb.12885. doi:10.1111/gcbb.12885

Ref. 41: Schmidt HP, Pandit BH, Cornelissen G, Kammann CI. Biochar-Based Fertilization with Liquid Nutrient Enrichment: 21 Field Trials Covering 13 Crop Species in Nepal. Land Degradation and Development. 2017;28: 2324–2342. doi:10.1002/ldr.2761 

149-156 7 Statement: Soil samples were taken in representative fields of interviewed farmers of three soil types to a depth of 20 cm (agricultural horizon). A random selection of sample spots were chosen and samples have then thoroughly been mixed to one composite sample for each soil type, air-dried and kept in sampling bags before transport and analysis. Sampling of the Ferralsol took place on a hillslope, whereas the Vertisol was taken directly from a field next to the riverbank (both in Gopalpura town). The Anthroposol represents a native Ferralsol where river (tank) sediments were applied [47] and was sampled in Berambadi town near the forest to the state border of Kerala.

Remarks: Please make this paragraph more precise and up to the mark with only relevant numbers and information.

We thank the reviewer for this comment. We have tried to condense and reformulate the paragraph so that it only contains the most important information regarding the sampling of soil, see lines 163 to 169 in the revised manuscript. 

199-200 9 Statement: The samples were preliminary stirred for 30 minutes and then pH was measured in the soil suspension.

Remarks: Please do mention the rate of stirring in rpm in 30 minutes 

We have added this information in the sentence in line 221 to 225 in the revised manuscript. Thanks for the suggestion.

315-317 14 Statement: We did not observe differences related to the biochar quality, probably because the differences between soils were not so important. 

Remarks: Please try to avoid such generalised statements or reframe it.

We thank the reviewer for this comment and rephrased it accordingly, more specifically, see lines 425 to 429 in the revised manuscript. 

351-354 15 Statement: Our empirical data from the interviews revealed that farmers used specific traditional practices like composting (97 % of farmers), ranging from traditional techniques like Jeevamrutha (liquid compost, 17 %) or farmyard manure preparation (97 %) to more knowledge-intensive setups of composting (i.e. vermicomposting, 28 %). 

Remark: Please mention the formulation or composition of Jeevamrutha used here. As this formulation changes a lot with the traditional knowledge of local people and local availability of resources.

We thank the reviewer for this suggestion and added the relevant information on the composition of Jeevamrutha in lines 482 to 486 in the revised manuscript. 

492-495 23 Statement: Future research could therefore provide answers to questions regarding type, quality and rates of OM applied, the influence of a changing climate on tropical agro-ecosystems as well as future coping strategies of rural farmer communities. 

Remarks: Please replace the word ‘coping’ with ‘cropping’ 

We thank the reviewer for this comment and changed the wording in this sentence accordingly (see line 642 to 645 in the revised manuscript).

Remarks: Please cite the below mentioned papers in Introduction part as recent literature review on biochar and agricultural biotechnology based new interventions. 

1. Maddalwar, S., Kumar Nayak, K., Kumar, M., & Singh, L. (2021). Plant microbial fuel cell: Opportunities, challenges, and prospects. Bioresource Technology, 341, 125772. https://doi.org/10.1016/j.biortech.2021.125772

2. Kumar, Manish, Shanta Dutta, Siming You, Gang Luo, Shicheng Zhang, Pau Loke Show, Ankush D. Sawarkar, Lal Singh, and Daniel CW Tsang. "A critical review on biochar for enhancing biogas production from anaerobic digestion of food waste and sludge." Journal of Cleaner Production (2021): 127143.

We thank the reviewer for the suggested references. We have added the second citation in the introduction (see lines 95 to 98 in the revised manuscript), but the first one appeared to us too far from the topic to be included.

---

## [Decision Letter · Decision Letter 1]

17 Jan 2022

Tailor-made biochar systems: Interdisciplinary evaluations of ecosystem services and farmer livelihoods in tropical agro-ecosystems

PONE-D-21-24354R1

Dear Dr. Abiven,

We’re pleased to inform you that your manuscript has been judged scientifically suitable for publication and will be formally accepted for publication once it meets all outstanding technical requirements.

Kind regards,

Primo Proietti

Academic Editor

PLOS ONE

Additional Editor Comments (optional):

Reviewers' comments:

Reviewer's Responses to Questions

**Comments to the Author**

1. If the authors have adequately addressed your comments raised in a previous round of review and you feel that this manuscript is now acceptable for publication, you may indicate that here to bypass the “Comments to the Author” section, enter your conflict of interest statement in the “Confidential to Editor” section, and submit your "Accept" recommendation.

Reviewer #1: All comments have been addressed

2. Is the manuscript technically sound, and do the data support the conclusions?

Reviewer #1: Yes

3. Has the statistical analysis been performed appropriately and rigorously? 

Reviewer #1: Yes

4. Have the authors made all data underlying the findings in their manuscript fully available?

Reviewer #1: Yes

5. Is the manuscript presented in an intelligible fashion and written in standard English?

Reviewer #1: Yes

6. Review Comments to the Author

Reviewer #1: Authors have performed the recommended changes. The authors hav also answered in edetail to all the required comments. Paper can be accepted.

7. PLOS authors have the option to publish the peer review history of their article (what does this mean?). If published, this will include your full peer review and any attached files.

Reviewer #1: No

---

## [Editor Report · Acceptance letter]

21 Jan 2022

PONE-D-21-24354R1 

Tailor-made biochar systems: Interdisciplinary evaluations of ecosystem services and farmer livelihoods in tropical agro-ecosystems 

Dear Dr. Abiven:

I'm pleased to inform you that your manuscript has been deemed suitable for publication in PLOS ONE. Congratulations! Your manuscript is now with our production department. 

Kind regards, 

on behalf of

Dr. Primo Proietti 

Academic Editor

PLOS ONE